# CD4^+^FOXP3^+^ T Cells in Rheumatoid Arthritis Bone Marrow Are Partially Impaired

**DOI:** 10.3390/cells9030549

**Published:** 2020-02-26

**Authors:** Magdalena Massalska, Anna Radzikowska, Ewa Kuca-Warnawin, Magdalena Plebanczyk, Monika Prochorec-Sobieszek, Urszula Skalska, Weronika Kurowska, Pawel Maldyk, Ewa Kontny, Hans-Jürgen Gober, Wlodzimierz Maslinski

**Affiliations:** 1Department of Pathophysiology and Immunology, National Institute of Geriatrics, Rheumatology, and Rehabilitation, 02-637 Warsaw, Poland; anna.radzikowska@spartanska.pl (A.R.); ewa.kuca-warnawin@spartanska.pl (E.K.-W.); magdalena.plebanczyk@spartanska.pl (M.P.); urszula.skalska@spartanska.pl (U.S.); weronika.kurowska@spartanska.pl (W.K.); ewa.kontny@spartanska.pl (E.K.); wlodzimierz.maslinski@spartanska.pl (W.M.); 2Department of Pathology, National Institute of Geriatrics, Rheumatology, and Rehabilitation, 02-637 Warsaw, Poland; monika.prochorec@interia.pl; 3Department of Diagnostic Hematology, Institute of Hematology and Transfusion Medicine, 02-776 Warsaw, Poland; 4Department of Rheumoorthopaedic Surgery, National Institute of Geriatrics, Rheumatology, and Rehabilitation, 02-637 Warsaw, Poland; klort@wum.edu.pl; 5Clinical Department of Orthopedic and Traumatology of Locomotor System, Enfant-Jesus Clinical Hospital, 02-005 Warsaw, Poland; 6Department of Pharmacy, Kepler University Hospital, 4020 Linz, Austria; Hans.Gober@bcchr.ca; 7Pharmaceutical Outcomes Programme, British Columbia Children’s Hospital, Vancouver, BC V5Z 4H4, Canada

**Keywords:** bone marrow, CD4^+^FOXP3^+^, Treg cells, rheumatoid arthritis, immunosuppression

## Abstract

There is evolving evidence that dysregulation of immune homeostasis in the bone marrow (BM) adjacent to the inflamed joints is involved in the pathogenesis of. In this study, we are addressing the phenotype and function of regulatory T cells (Tregs) residing in the BM of patients with rheumatoid arthritis (RA) and osteoarthritis (OA). BM and peripheral blood samples were obtained from RA and OA patients undergoing hip replacement surgery. The number and phenotype of Tregs were analyzed by flow cytometry and immunohistochemistry. The function of Tregs was investigated ex vivo, addressing their suppressive activity on effector T cells. [^3^H]-Thymidine incorporation assay and specific enzyme-linked immunosorbent assay were used for quantification of cell proliferation and pro-inflammatory (TNF, IFN-γ) cytokine release, respectively. Significantly lower numbers of CD4^+^FOXP3^+^ T cells were found in the BM of patients with RA compared to control patients with OA. High expression of CD127 (IL-7α receptor) and relatively low expression of CXCR4 (receptor for stromal cell-derived factor CXCL12) are characteristics of the CD4^+^FOXP3^+^ cells residing in the BM of RA patients. The BM-resident Tregs of RA patients demonstrated a limited suppressive activity on the investigated immune response. Our results indicate that the reduced number and impaired functional properties of CD4^+^FOXP3^+^ T cells present in the BM of RA patients may favor the inflammatory process, which is observed in RA BM.

## 1. Introduction

Rheumatoid arthritis (RA) is an autoimmune disease, where chronic inflammation leads to joint destruction. Activated T and B lymphocytes, monocytes, fibroblast-like-synoviocytes and granulocytes are observed among massive cell infiltration into joint fluid, all of them producing mediators of inflammation and pro-inflammatory cytokines, contributing to initiation and perpetuation of joint destruction.

Regulatory T cells (Tregs) are a small subpopulation of T cells establishing tolerance and proper functionality of the immune system. Suppressive activity of Tregs defined as CD4^+^CD25^+^FOXP3^+^ is one of the main mechanisms controlling autoimmunity in animals and humans. Tregs can be classified into two main populations—thymus-derived Tregs (tTregs) and induced from CD25^−^ precursors in peripheral lymphoid organs, peripherally-induced Tregs (pTregs) [1]. Their suppressive function is accomplished through the release of cytokines (IL-10 or TGF-β) or cell-to-cell interactions. The molecular definition of natural Treg is based on the expression of transcription factor Foxp3 [2,3] and the presence of Treg in humans has been well documented in peripheral blood, umbilical cord blood and lymph nodes. Qualitative, as well as quantitative deficiencies of Treg, are proved to contribute to the development of the autoimmune pathology in many diseases [4]. The presence of Treg in inflamed synovial membrane and the enrichment of functional CD25^bright^CD4^+^ T cells in synovial fluid as compared to peripheral blood, demonstrated in different rheumatic diseases [5,6,7], suggest attempts of the immune system to limit ongoing autoimmune processes. Nowadays adoptive transfer of regulatory T cells is of great interest [8] and clinical trials with Tregs have been conducted in autoimmune disorders [9] and transplantation [10,11].

Since the last few years, several findings have strengthened the role of bone marrow in the development of adaptive immunity in the context of memory T cells [12,13]. Bone marrow (BM) is believed to maintain memory B cells as well as T cells, however there is no consensus about proliferation, activation and migratory capacity of BM T cells [14,15]. From one point of view, it is proved that BM memory T cells are resting in terms of proliferation and transcription [16,17] and express genes reported for tissue-resident memory T lymphocytes [18], while from the other, memory T cells strongly divide in BM and recirculate in and out the BM [19]. It is probable that there are two distinct BM niches supporting two distinguished features of memory T cells—namely recirculation in and out the BM and resting [12]. BM, under certain conditions, can serve as a secondary lymphoid organ, where antigen presentation, even higher than in lymph nodes, occurs [20]. In this context, it is not surprising that BM can serve as a reservoir of the inflammatory T cells, as was found by the transfer of chronic colitis to severe combined immunodeficiency (SCID) mice by BM memory CD4^+^ cells [21], but also as a source of CD4^+^CD25^+^ Tregs [22]. These observations further support the notion that BM can be considered as a lymphoid organ contributing to T-cell immunity and homeostasis as well. Our group has previously demonstrated that bone marrow actively participates in RA pathogenesis by Toll-like receptors (TLR) triggered B cell activation [23], presence of a high number of activated T cells [24] and development of Th17 cell responses [25].

In this work, we have verified the hypothesis that CD4^+^FOXP3^+^ cells are present and functionally active in the BM of RA and OA patients. The phenotype of BM-resident CD4^+^FOXP3^+^ T cells of RA patients appeared to be different from the phenotype of Tregs isolated from peripheral blood of the same patient. We found that Tregs present in RA BM are mostly memory cells expressing a high level of CD45RO, a rather low level of CD25 and a low level of chemokine receptor CXCR4, which may be responsible for reduced retention of this population in RA BM. They also demonstrated a limited suppressive activity on investigated immune response, which, together with their reduced number, may favor the inflammatory process described in the bone marrow of RA patients. 

## 2. Materials and Methods

### 2.1. Patients

Two groups of patients were enrolled in the study: RA and OA patients. Femoral BM samples were obtained from patients during complete hip bone replacement surgery as a standard procedure. Additionally, peripheral blood (20 mL) was obtained from six RA and six OA patients before surgery. All patients fulfilled the American College of Rheumatology’s revised criteria for RA [26] and OA [27]. The RA patient group was composed of 36 patients and the OA patient group was composed of 42 patients. All participants gave written informed consent according to the Declaration of Helsinki, and the study was approved by the Institute of Rheumatology (now the National Institute of Geriatrics, Rheumatology, and Rehabilitation) Ethics Committee in Warsaw (reference number 2011/03/31). All clinical data concerning patients are summarized in Table 1.

### 2.2. Cell Isolation, Gating Strategy and Phenotypic Analysis

Human peripheral blood mononuclear cells (PBMCs) were isolated from freshly drawn venous blood and BM mononuclear cells (BMMCs) were obtained from BM in both of OA and RA patients by Ficoll-Paque PLUS (GE Healthcare Biosciences AB, Uppsala, Sweden) density centrifugation. To examine the phenotype of regulatory T cells (Tregs), the following antibodies were used: anti-CD4-PerCP, anti-CD25-FITC, anti-CD127-PE, anti-CD45RA-FITC, anti-CD45RO-PE and anti-CD184 (CXCR4)-PE (all from BD Biosciences, San Jose, CA, USA). Next, cells were fixed and permeabilized using the Foxp3 Staining Buffer Set followed by intracellular staining with anti-Foxp3-APC antibody (PCH101; eBioscience, San Diego, CA, USA). Cells were acquired using FACSCalibur (BD Biosciences), and the results were analyzed using CellQuest (BD Biosciences) software. The gating strategy was based on the identification of lymphocytes according to FSC and SSC signal distribution. Then, CD4^+^ lymphocytes were gated, and next, appropriate cell populations were identified. Gates were settled according to the isotype control or fluorescence minus one (FMO) for the desired marker. The dot plots shown are representative for at least six experiments done on different patients. The described gating strategy is shown in Figure 2b and was used though all the analysis of the cell phenotype presented in this paper.

### 2.3. Cell Separation

BMMCs or PBMCs were cultured for 48 h in 75 cm^2^ culture flasks (50 × 10^6^ cells/flask/20 mL) (Nunc, Roskilde, Denmark) in RPMI-1640 (Invitrogen, Paisley, UK), supplemented with l-glutamine (2 mM), HEPES (10 mM), penicillin (100 IU/mL), streptomycin (100 μg/mL), kanamycin (100 μg/mL), plasmocin (25 μg/mL; InvivoGen, San Diego, CA, USA) and 10% fetal calf serum (FCS) (Biochrom AG, Berlin, Germany) before the separation of Tregs and responder T cells (Tresps). First, CD4^+^ T cells were enriched using a MiniMACS device (Miltenyi Biotec, Bergisch Gladbach, Germany). Then, cells stained with anti-CD4-FITC and anti-CD25-PE antibodies were incubated with 7-amino-actinomycin D (7-AAD; 5 μL, 10 min, room temperature) to stain and exclude dead cells. CD4^+^CD25^−^ Tresp and CD4^+^CD25^++^ Treg populations were sorted using FACSAria (BD Biosciences) according to the expression of CD25. Starting from at least 100 × 10^6^ BMMCs, isolated from BM sample obtained directly after surgery, we usually got less than 80,000 Tregs. Analysis and sorting gates were restricted to single-live-lymphocytes by using their light scattering characteristics, forward scatter and side scatter and live cells (7-AAD negative). The described gating strategy is shown in Figure 6a and was used in all the sorting experiments presented in this paper. Sorted populations were counted and used for functional assays, and to verify the purity of isolated cells. The purity of isolated cells was estimated in the context of CD25 and FOXP3 expression (here sorted cells were additionally stained intracellularly for FOXP3).

### 2.4. Functional Assay

For suppression assays, CD4^+^CD25^−^ and CD4^+^CD25^++^ T cells were isolated from the BM of RA and OA patients and from the peripheral blood of healthy blood donors. Sorted CD4^+^CD25^++^ Tregs were gated by selecting CD4^+^ cells with the brightest CD25 expression (the highest one-third of CD25 expression) and sorted Tresps were gated by selecting CD4^+^ cells with no CD25 expression (according to the isotype control). Isolated Treg and Tresp populations were cultured in triplicates in U-bottom 96-well plates (Nunc, Roskilde, Denmark) for 5 days in a complete RPMI-1640 medium supplemented with 10% FCS as described previously. The indicated number of CD4^+^CD25^−^ and CD4^+^CD25^++^ (1–3 × 10^3^/well) were cultured separately or in a 1:1 ratio, together with a 10-fold excess of T-cell-depleted accessory cells [28]. Using different rates of Treg:Tresp, useful while investigating Treg function, were not possible because of the very small number of pure Tregs population isolated from human bone marrow. T-cell-depleted accessory cells were isolated by the negative selection of allogenic PBMCs from healthy volunteers stained with anti-CD3-FITC mAb and sorted by using FACSAria (BD Biosciences), followed by mitomycin C (MitC; Sigma-Aldrich, St. Louis, MO, USA) treatment to arrest cells in the cell cycle. When indicated, cells were additionally stimulated with soluble anti-CD3 (clone Hit3a, 2.5 μg/mL) and anti-CD28 (clone CD28.2, 2.5 μg/mL; both from BD Pharmingen, San Diego, CA, USA) [29].

Proliferation of T cells was assessed by measuring the incorporation of [^3^H]-thymidine added for the final 18 h before harvesting ([^3^H]-TdR (radiolabeled thymidine); 0.5 μCi/well; GE Healthcare Ltd, Buckinghamshire, UK) [28]. Just before the addition of [^3^H]-thymidine, one-half of the culture supernatant (100 μL) was removed from each well, centrifuged and frozen in −70 °C until specific enzyme-linked immunosorbent assays (ELISAs) were performed. Each experiment was done at least six times using cells isolated from different BM donors and different peripheral blood healthy volunteers.

To investigate the suppression potential of Tregs, proliferation (or cytokine production) of Tresps and Tregs cultured together was compared with proliferation (or cytokine production) of Tresps cultured alone. Statistical significance was calculated comparing real values (cpm in case of proliferation and pg/mL in case of cytokines production).

### 2.5. Evaluation of Cytokine Production

The concentration of tested cytokines (TNF, IFN-γ, IL-10, IL-35 and TGF-β) was detected in culture supernatants using a specific ELISA. The secretion of TNF was evaluated using Human TNF DuoSet (R&D Systems, Minneapolis, MN, USA); that of IFN γ, IL-10 and TGF-β by Human ELISA Ready-SET-Go!® (eBioscience); and that of IL-35 concentration by ELISA Kit for IL-35 (Uscn Life Science Inc., Wuhan, People’s Republic of China). Optical density was determined with an ELISA reader at 450 nm, and the detection range was 15.6–1000 pg/mL for TNF, IL-35 and TGF-β, whereas it was 3.6–500 pg/mL for IFN-γ and 4.4–300 pg/mL for IL-10.

### 2.6. Immunohistochemistry

BM biopsies obtained from orthopedic surgery were examined histopathologically. They were fixed in Oxford fixative (formaldehyde 40%, glacial acetic acid, sodium chloride, distilled water), routinely processed and embedded in paraffin wax. Sections of 3 μm thick were cut and stained with hematoxylin and eosin. The following antibodies were used: CD3 (polyclonal Ab, dilution 1:50; Dako, Glostrup, Denmark), CD4 (clone 4B12, dilution 1:10; Novocastra, now part of Leica Microsystems, Wetzlar, Germany), Foxp3 (clone 22510, dilution 1:50; Abcam, Cambridge, UK). Staining was performed according to the manufacturer’s instructions. The EnVision Detection System (Dako Denmark A/S) was used for detection. Positive controls were performed using human tonsils. Negative (isotype) controls were performed using ready-to-use FLEX negative control mouse antibodies (cocktail of mouse IgG1, IgG2a, IgG2b, IgG3 and IgM; code nr IR750; Dako Denmark A/S). Samples were reviewed for expression of these proteins in a blinded study performed by a research scientist (MP-S). Appropriate cellular localization for immunostaining was membrane −CD3, −CD4 and nuclear −Foxp3. All photographs were taken using Olympus microscope cameras: DP72 Olympus BX63 and DP12 Olympus BX (Olympus, Tokyo, Japan).

### 2.7. Statistical Analysis

D’Agostino–Pearson normality test was used to confirm the use of parametric or non-parametric tests for further analysis. Differences between cell populations in BM in groups of patients (OA vs. RA) were analyzed by the two-tailed Mann–Whitney U-test and presented in figures as median ± min/max. Comparison of BM with blood from the same patients (done separately for OA and RA patients) were analyzed by the Wilcoxon test while the comparison of BM or blood between OA and RA patient groups were analyzed by Mann–Whitney U-test. The differences in suppression potential between the groups of patients were tested for their statistical significance using the parametric two-tailed T-test or the Wilcoxon test when the normality assumption was not met (GraphPad Software, USA). Data were analyzed using Statistica (version 6.0) and Prism 4.0 (GraphPad, La Jolla, USA) software. For all tests, a value of *p* < 0.05 was considered significant. 

## 3. Results

### 3.1. FOXP3^+^ T Cells Are Present in the BM of Patients RA

Histopathological examination of BM biopsies exhibited the presence of FOXP3^+^ positive cells among CD3^+^ and CD4^+^ lymphocytes in the BM obtained from RA and OA patients (Figure 1a–h). In order to quantify and analyze the phenotype of CD4^+^FOXP3^+^ cells in the BM of OA and RA patients, the BMMCs were isolated from both patient groups, and the phenotype of Tregs was examined by FACS analysis.

### 3.2. Proportions of CD4^+^FOXP3^+^ T Cells Are Lower in RA than in OA BM

The proportion of CD4^+^FOXP3^+^ cells among the CD4^+^ population was significantly lower in the BM of RA in comparison with OA patients (Figure 2a,b), although the level of FOXP3 expression per cell in both patient groups was similar. Representative dot plots showing FACS analysis of FOXP3 distribution on gated CD4^+^ T cells are presented in Figure 2b.

To determine the potential differences in CD4^+^FOXP3^+^ pool composition between the peripheral blood and the BM, we compared the populations of potential Tregs within PBMCs and BMMCs isolated from the same patient. Surface expression of CD25 was discovered as the first marker of potential Tregs, many years before Foxp3 had been identified as the main transcription factor responsible for Treg phenotype [2]. However, we found a significantly lower proportion of CD4^+^CD25^+^ as well as CD25^+^FOXP3^+^ cells in the BM in comparison with the peripheral blood in both OA and RA patient groups (Figure 2c,d). Although patients were treated with different drugs, we did not observe any significant differences in the CD4^+^FOXP3^+^ number depending on the kinds of drugs taken.

### 3.3. Low Expression of CXCR4 Is Observed in RA BM CD4^+^FOXP3^+^ Cells

To evaluate whether CD4^+^FOXP3^+^ cells have the potential to migrate into and out the BM, we investigated their chemokine receptor CXCR4 expression that is fundamental for the recruitment of hematopoietic stem cell into the BM [19,22]. We found a significantly lower proportion of CD4^+^ T cells expressing CXCR4 in BM isolated from RA patients, in comparison with OA patients (Figure 3a,b). We also observed the lower proportion of CD4^+^FOXP3^+^ Tregs expressing CXCR4 in the BM and blood from RA patients in comparison with OA patients (Figure 3c,d), which may be at least partially responsible for their reduced trafficking to the BM and for the presence of decreased proportion of Tregs in the BM of RA patients.

### 3.4. CD4^+^FOXP3^+^ Cells from RA BM Are Mostly CD127^+^

CD127 is the receptor for IL-7, a cytokine that is produced by a variety of stromal cells and thus present in bone marrow [24]. Since the most reliable cell surface markers used for isolation of live CD4^+^FOXP3^+^ Tregs were based on the high expression of CD25 and the low expression of CD127 [30,31], we examined BM cells for the expression level of these markers. Most of the CD4^+^ T cells isolated from the BM of OA and RA patients expressed CD127 (75.7% ± 2.0% and 77.8% ± 1.7% of OA and RA CD4^+^ T cells, respectively) with a similar level of CD127 expression per cell. We found fewer CD127^−^CD25^+^ T cells in RA BM compared to OA BM, although the difference between the groups did not reach the statistical significance (1.19 ± 0.36 in OA vs. 0.67 ± 0.28 in RA; *p* = 0.423), and only a tendency was observed (Figure 4a). However, when examining their distribution in the individual patients, we found a significantly lower percentage of CD127^−^CD25^+^ T cells in the BM compared to the peripheral blood (Figure 4a,b).

In the BM of RA patients, a high percentage of CD4^+^FOXP3^+^ cells expressed CD127 (Figure 4c,d). This CD4^+^FOXP3^+^CD127^+^ T-cell population was significantly enlarged in the BM of RA patients compared to the BM of OA patients and to the peripheral blood in RA patients (Figure 4d). In contrast, in OA patients, this population was of similar size in the BM as well as the peripheral blood (Figure 4d, last graph). The increased number of CD4^+^FOXP3^+^CD127^+^ cells present in the BM of RA patients may suggest an activation process occurring in RA BM, as studies on wild-type mice found that CD127 is highly expressed on activated Tregs, in both lymphoid and non-lymphoid tissues [32].

To address the question, how many cells gated as CD127^−^CD25^+^ in BM express FOXP3 and whether such a gating strategy is meaningful in terms of the isolation of Tregs from the BM, we followed the methodology of analysis described previously [31]. The relationship between CD4, CD127, CD25, and FOXP3 revealed that, indeed, gating on CD4^+^CD127^−^CD25^+^ cells represents the most numerous group of CD4^+^FOXP3^+^ cells in the peripheral blood of both OA and RA patients, as well as in the BM of OA patients (more than 40%). However, in the BM of RA patients, less than 10% of CD4^+^CD127^−^CD25^+^ cells were expressing FOXP3, making described sorting strategy not useful in case of bone marrow as a source of CD4^+^FOXP3^+^ cells (Table 2).

### 3.5. CD4^+^FOXP3^+^ Cells from RA BM Are CD45RO^+^ Memory Cells

In order to verify the proportions of naive and memory Tregs, we investigated CD45RA and CD45RO expression on CD4^+^FOXP3^+^ Tregs in the BM and peripheral blood of the same patient. Regardless of the disease diagnosis (OA and RA) or the investigated tissue (BM and peripheral blood), most CD4^+^FOXP3^+^ Tregs expressed the CD45RO^+^ memory phenotype, which is in agreement with other studies (Figure 5) [33].

### 3.6. CD4^+^FOXP3^+^ Cells in the BM of RA Patients Demonstrate Limited Suppressive Potential

To test the suppressive ability of the Tregs isolated from BM, CD4^+^CD25^++^ Tregs were cultured with CD4^+^CD25^−^ Tresps, isolated from the BM of the same patient. The purity of sorted populations was more than 96% (Figure 6a). More than 90% of isolated CD4^+^CD25^++^ cells showed FOXP3 expression, whereas more than 97% of CD4^+^CD25^−^ responder cells were FOXP3 negative (Appendix A).

We found that responder T cells isolated from healthy blood donors proliferated more vigorously in comparison to the same population isolated from the bone marrow of RA patients (9192.1 ± 1970.0 vs. 981.2 ± 489.8; *p* = 0.0005) or OA patients (9192.1 ± 1970,0 vs. 5049.2 ± 3676.6; *p* = 0.03). BM responder T cells were not able to proliferate so firmly as Tresps from peripheral blood even after additional stimulation by anti-CD3/CD28 antibodies (42724.2 ± 10993.7 vs. 6093.3 ± 4547.6; *p* = 0.002; healthy blood vs. RA BM). Proliferation of stimulated Tresp from OA BM, although higher than from RA BM, was much lower than that obtained from Tresp from healthy blood donors (11777.1 ± 5568.6 vs. 42724.2 ± 10993.7; ns). The impaired proliferation of BM effector T cells has probably made impossible the real investigation of BM Tregs suppressive activity. Regulatory T cells isolated from blood or BM were anergic and did not respond for anti-CD3/CD28 stimulation. CD4^+^FOXP3^+^ cells isolated from both OA and RA BM were not able to significantly inhibit proliferation of Tresps, counted as cpm comparison between Tregs and (Tregs plus Tresp) coculture. Tregs population isolated from the blood of healthy donors were suppressive (Figure 6b), but conditions of elevated stimulation via anti-CD3/CD28 antibodies, abolished the observed effect.

Next, the effect of CD4^+^CD25^++^ Tregs on cytokines secretion in coculture with CD4^+^CD25^−^ Tresps was investigated. For this purpose, we analyzed cell culture supernatants from cultures depicted in Figure 6a for levels of TNF, IFN-γ, IL-10, IL-35 and TGF-β.

TNF holds the central place in the pathogenesis of many autoimmune diseases, and its participation in the process leading to RA is well documented [34]. On the contrary, IFN-γ considered mainly as an inflammatory mediator, appeared also to be important for Treg functions [35]. Tregs and Tresps isolated from BM of RA patients produced similar amounts of TNF and no suppression exerted by Tregs on TNF production was observed. Significant suppression of TNF production (Figure 5c) was detected only in cultures of Tresps isolated from peripheral blood of healthy donors when cocultured with their corresponding Tregs. The levels of TNF and IFN-γ secretion mirrored levels of observed proliferation when additional stimulation (anti-CD3/CD28) was present. 

There are few cytokines (IL-10, TGF-β and IL-35) that may be produced by Tregs and potentially involved in the mechanism of suppression exerted by those cells. The amount of IL-10 produced by BM Tregs and Tresps was negligible, and IL-35 production was not detected at all (results not shown). As shown in Figure 5e, the TGF-β amount produced by Tregs and responder T cells isolated from BM from OA and RA patients as well as from healthy blood donors were similar. Additional stimulation by anti-CD3/CD28 antibodies did not have any impact on the observed result.

Collectively, these results support the suppressive potential of CD4^+^CD25^++^ Tregs isolated from peripheral blood from healthy blood donors, but not Tregs residing in the BM of RA and OA patients. These data, however, cannot exclude the functionality of BM Tregs, which was hard to prove because of the weak proliferation of responder T cells.

## 4. Discussion

There is growing evidence of published results supporting the concept that bone marrow takes an active part in the induction of immunological response [20,36]. It is believed that the presence of inflammatory, memory CD4^+^ and plasma B cells resistant for conventional therapy in BM may cause the reason why chronic diseases, including rheumatic diseases, are still incurable today [37]. Although symptoms of inflammation may be alleviated, disease progression eventually mitigated and physical function improved however, continuous treatment with anti-inflammatory medication is required in the majority of patients even including biological DMARDs.

The BM-centered disease model of RA pathogenesis points to the BM as the source of potentially pathologic cells that migrate to synovium through bone canaliculi or exert their functions from the subchondral side of the bone [38]. In RA patients—independently of the ongoing process of synovial tissue inflammation—the presence of subchondral BM inflammation (BM edema) is observed [38], which correlates with clinical signs of RA and is considered as a predictor of rapid radiological progression of the disease [39]. However, it is possible, that bone marrow may actively participate not only in the inflammatory process in RA [23,24], but also in tolerance induction and maintenance supporting Treg development. Recent data point to the role of bone marrow T cells in the immune system. BM Tregs seem to be required for optimal control of GVHD post-transplant [40] while redirection to the BM was shown to increase T cell antitumor functions [41].

The main regulators of T cell homeostasis and immune tolerance are CD4^+^CD25^+^ tTregs that originate from the thymus and express the transcription factor FOXP3^+^ [42]. Human Treg cells constitute 1–2% of CD4^+^ T cells in human peripheral blood and their phenotype is well studied [28]. However, not much is known about Tregs in human BM [22,43] and the results are not consistent [22,36]. Zou [22] showed that a bigger population of CD4^+^CD25^+^ cells is present in BM than in blood, lymph nodes or thymus of healthy people, but noteworthy, the authors investigated cells mobilized from bone marrow to blood by granulocyte colony-stimulating factor (G-CSF). On contrary, Wang et al. presented significantly decreased Treg numbers in BM of RA patients compared with the matched peripheral blood of RA patients or OA BM and positive correlations of each T subpopulation between BM and peripheral blood samples from RA patients [36].

In this report, we could demonstrate that significantly smaller proportions of CD4^+^FOXP3^+^ cells are present in RA BM in comparison with OA BM and that there are significantly fewer Tregs in BM as compared to blood from the same patient (in OA and RA patient groups). This insufficiency may favor the development of autoimmunity, although it is not known whether it is a primary or secondary effect in the disease pathogenesis. Interestingly, we also noticed significantly fewer CD25^−^FOXP3^+^ cells in RA than OA bone marrow. It was shown before that Foxp3 expression and not CD25 directly correlates with Treg suppressive properties and CD4^+^CD25^lo^Foxp3^+^ cells exert similar suppressive properties as CD4^+^CD25^hi^Foxp3^+^ [44]. Thus, the observed CD4^+^CD25^−^FOXP3^+^ population in BM may represent regulatory T cells, which was already described in patients with SLE [45]. CD4^+^CD25^−^Foxp3^+^ cell population may also represent uncommitted Treg cells with labile Foxp3 expression [46]. Because of its proximity to inflamed/destroyed joint in RA, Treg cells may leave the bone marrow and migrate toward inflamed joints, causing decreased Treg number detected in BM. It was proved, that under arthritis conditions CD4^+^CD25^lo^Foxp3^+^ cells may lose Foxp3 expression and undergo trans-differentiation into Th17 cells, that accumulate in inflamed joints and participate in the pathogenesis of autoimmune arthritis which was shown on mice [47] and human [48] cells. This process may also progress in the bone marrow of RA patients, finally resulting in a lower number of FOXP3^+^ cells detected in this compartment.

Another important factor that might have influenced the Treg population is the kind of drugs that patients were taking before the operation. We have analyzed sixteen OA patients in the context of the percentage of CD4^+^FOXP3^+^ cells in freshly isolated BMMC. In this group five patients were taking NSAIDs, six were taking other drugs lowering the pain but not-NSAID and five patients were declaring not taking any drugs. NSAIDs are a heterogenic group of drugs, suppressing the production of the enzyme cyclooxygenase (COX), whose product is prostaglandin E2 (PGE2). Data concerning the prostaglandin effect on FOXP3 are not consistent. On the one hand, it was shown that PGE2 upregulates both mRNA and protein expression of Foxp3, induces the regulatory phenotype in CD4^+^CD25^−^ cells and increases suppressive functions of isolated human CD4^+^CD25^+++^ Tregs [49]. On the other, PGE2 can inhibit the differentiation of Treg mediated through the EP2-cAMP/PKA signaling pathway [50]. In our work, we did not observe any significant differences in CD4^+^FOXP3^+^ numbers depending on the kind of drugs taken in the OA patients’ group.

In the RA patients’ group, there were patients treated with methotrexate or glucocorticoids. Comparing the percentage of CD4^+^FOXP3^+^ cells in the group of patients taking methotrexate (*n* = 8) and not taking methotrexate (*n* = 8), we did not observe any differences, which is consistent with present knowledge that methotrexate does not influence the Treg functions [51]. We noted different percent of Treg when we compared RA patients taking two different glucocorticoids: methylprednisolone (*n* = 8) and encorton (*n* = 5), but the differences were not statistically significant. Treatment of each steroid was stopped at least one week before surgery.

In vitro studies indicate that glucocorticoids (GC) favor the expansion of activated Treg cells which are saved from apoptosis and stimulate differentiation of CD4^+^ cells into pTreg [52]. On the other side, GC determine the apoptosis of resting Tregs. In vivo studies support the notion that GC-induced Treg expansion is dependent on the activation status of Treg, a kind of disease and possibly on the tissue. Because most of the investigated patients were treated by steroids, we could suspect the increased number of Treg in RA BM. However, the observed lower Treg number in RA BM in comparison to OA BM may either indicate that Treg in BM are not activated and thus prone to apoptosis mediated by GC or that published before observations are not valid for BM.

Our data do not give information on the origin of Tregs in BM. Probably they migrate from the thymus, but the local development in BM in a thymus independent process from CD4^+^CD25^−^ cells cannot be ruled out. As it was shown in many studies, human Tregs accumulate in the joins with the ongoing inflammatory process [6,7,40,46] and are fully functional there [53,54]. Independent of the diagnosis (psoriatic arthritis, spondyloarthropathy, juvenile idiopathic arthritis, RA) or the applied treatment, the Treg population is constantly higher in joints than in the blood, which suggests an active process of Treg recruitment from the blood to the site of inflammation [5]. Enlarged bone canals in RA may allow migration of cells directly from BM to the joints [55,56].

The main factor responsible for the migration of cells toward BM is stromal cell-derived factor-1 (SDF-1, CXCL12). The sole receptor for CXCL12 is CXCR4 (CD184) present on T and B lymphocytes, dendritic cells, monocytes and endothelial cells [57]. As memory cells respond better to CXCL12 than naive [58] and regulatory T cells present in the bone marrow are mostly memory cells, they were expected to be good responders to CXCL12. However, presented here, a lower expression of CXCR4 on CD4^+^ T cells and CD4^+^FOXP3^+^ cells from RA BM in comparison to cells isolated from OA BM may be responsible for weaker retention of Treg inside the bone marrow niches in RA patients.

Although FOXP3^+^ Tregs are characterized by low expression of CD127 in addition to high expression of CD25 [30,31], all human CD4^+^ and CD8^+^ cells may exert transient expression of CD25 and FOXP3 during activation [59]. Activation results also in the down-regulation of CD127 (IL-7Rα) expression on all human T cells [60]. Because a similar concentration of IL-7 was detected in OA and RA BM [24], higher expression of CD127 observed on Tregs from RA BM and presented here (Figure 4) might underline the role of IL-7 in Treg development in RA BM. CD127 might be expressed on Tregs to serve a function such as local expansion, restoration of suppressive activity, retention of Tregs in the BM or an anti-apoptotic effect. BM mesenchymal stroma is known to be the abundant source for IL-7 and thus can serve as a place for maintenance or expansion of Tregs, provided they express CD127. Alternatively, CD127^+^ Tregs may be generated in the periphery and captured in the BM.

As medical treatment of the investigated RA patients varied (NSAIDs, methotrexate), but overall results including CD127 expression by T cells were comparable, it is unlikely that any of those drugs could be responsible for the observed high expression of CD127 on BM CD4^+^FOXP3^+^ cells. Interestingly, Tregs in the synovial fluid of patients with RA do not express CD127 [61]; however, it cannot be ruled out that Tregs from the BM lose CD127 during migration to the joint or in subsequent steps of development. Importantly, previous studies suggest that Tregs need to downregulate CD127 for proper function. Although it was shown that IL-7 increases the number of Treg cells by inducing the peripheral expansion of thymic-independent Tregs in mice [62] and humans [61], IL-7 present in the arthritic joint can abrogate the suppression mediated by Tregs isolated from the synovial fluid of RA patients [61]. It is tempting to speculate that the modification of the functionality of Tregs may occur as a result of the inflammatory environment of the BM in RA patients. The role of IL-7 and CD127 on Tregs from BM would need further investigation to be fully understood.

CD4^+^CD25^+^ Tregs isolated from the peripheral blood of RA patients are phenotypically similar to those isolated from healthy donors and are suppressive in vitro [6,53]. In addition, Tregs accumulating in the joins with the ongoing inflammatory processes are fully functional, as it was shown in several studies [53,54]. However, it was reported that although peripheral Tregs in RA patients are capable of suppressing the proliferation of effector T cells, they were unable to suppress the production of pro-inflammatory cytokines (TNF, IFN-γ) by monocytes and activated T lymphocytes, whereas therapy with anti-TNF immunoglobulins completely restored the Treg function [63]. Ehrenstein and others have shown that lack of suppression exerted by Tregs isolated from RA blood was not due to effector T-cell resistance, but due to the functional incapability of Tregs resulting from the presence of TNF [63]. In our experiments, as shown in Figure 6, Tregs isolated from the BM of both RA and OA patients neither were able to suppress Tresps proliferation nor TNF and IFN-γ production by Tresps. We have shown that Tregs and Tresps from RA BM produce high amounts of TNF (Figure 6c), which, together with a high concentration of TNF detected in bone marrow plasma of RA patients [25], may be responsible for the block of Tregs suppressive function in BM. Suppressive function of Tregs from healthy blood was affected by additional anti-CD3/CD28 stimulation in our experiments, which is consistent with previously shown data that Treg function inversely correlates with the activation status of effector T cells [6,29,64]. Counts below 10,000 cpm may reflect the diminished ability of BM cells, especially from RA, to proliferate even after strong stimulation (alloantigen and anti-CD3/CD28) in vitro. Despite this defect, the production of cytokines, especially TNF and TGF-β, are comparable to that exerted by cells isolated from blood.

Regulatory capacity of tTregs is identified in most assays to be contact-dependent, and soluble factors such as IL-10, TGF-β and IL-35 do not seem to be essential for the suppressive function in the peripheral tTreg subpopulation [65]. Although IL-10 from Tregs is critical for the maintenance of hematopoietic stem cell niche in the BM as an immune-privileged place in mice [66], we were able to detect only negligible amounts of IL-10 in BM Treg cultures, which is consistent with the results obtained by others investigating the properties of Tregs isolated from the peripheral blood [6,29]. IL-35 was not detected at any of our samples. Comparable concentrations of TGF-β produced by Treg and Tresp populations in the BM may suggest that this cytokine is not specifically involved in the suppression process (Figure 6e). However, we did not investigate the cell surface-bound TGF-β that was previously shown to participate in cell contact-dependent immunosuppression by CD4^+^CD25^+^ Tregs [67]. Importantly, TGF-β plays a major role in the maintenance of homeostasis of articular cartilage and subchondral bone [68] and its production in BM microenvironment is required for the maintenance and regulation of hematopoiesis [69]. TGF-β derived from Tregs and Tresps in the BM may participate in all these processes.

The presence of CD4^+^FOXP3^+^ Treg population in the BM of RA patients is an indication that the BM is a place of immune regulation. The BM may serve as a source of memory Tregs migrating toward inflamed joints in RA patients. However, diminished proportion and compromised activity of Tregs in the BM of RA patients might not be sufficient to control the ongoing inflammation in vivo and at least partially explain the process of ongoing inflammation.

## Figures and Tables

**Figure 1 cells-09-00549-f001:**
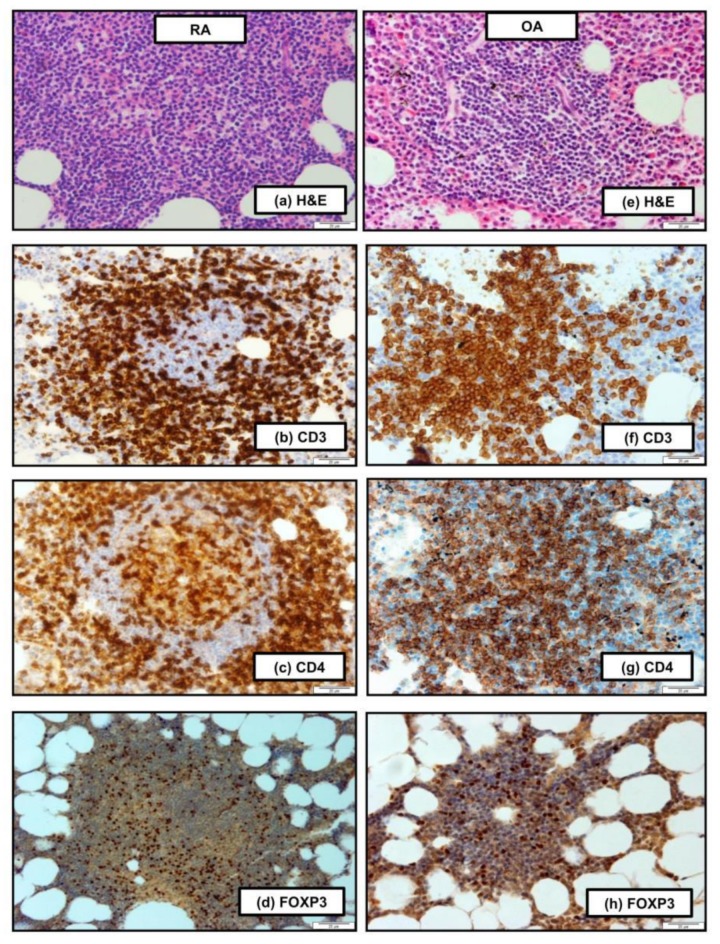
Histopathological features of the bone marrow (BM) of patients with rheumatoid arthritis (RA) (**a**–**d**) and osteoarthritis (OA) (**e**–**h**). (a) Nodular lymphocytic infiltration with germinal center formation (hematoxylin and eosin [H&E] stain, 100×). (b) CD3^+^ T cells in the marginal and mantle zone. (**c**) CD4^+^ T cells in the lymphoid follicle. (d) Nuclear expression of FOXP3 in cells localized in the lymphoid follicle. (b–d: EnVision stain, 100×). (e) H&E staining shows visible nodular lymphocytic infiltration, 100×. (f,g) Most of the lymphocytes in the lymphoid follicle revealed CD3 and CD4 expression. (**h**) FOXP3 in nuclear localization in cells of the lymphoid follicle (**f**–**h**: EnVision stain, 100−). Scale bar, 20 μm. Histology staining was done on five patients in each group while one representative is shown.

**Figure 2 cells-09-00549-f002:**
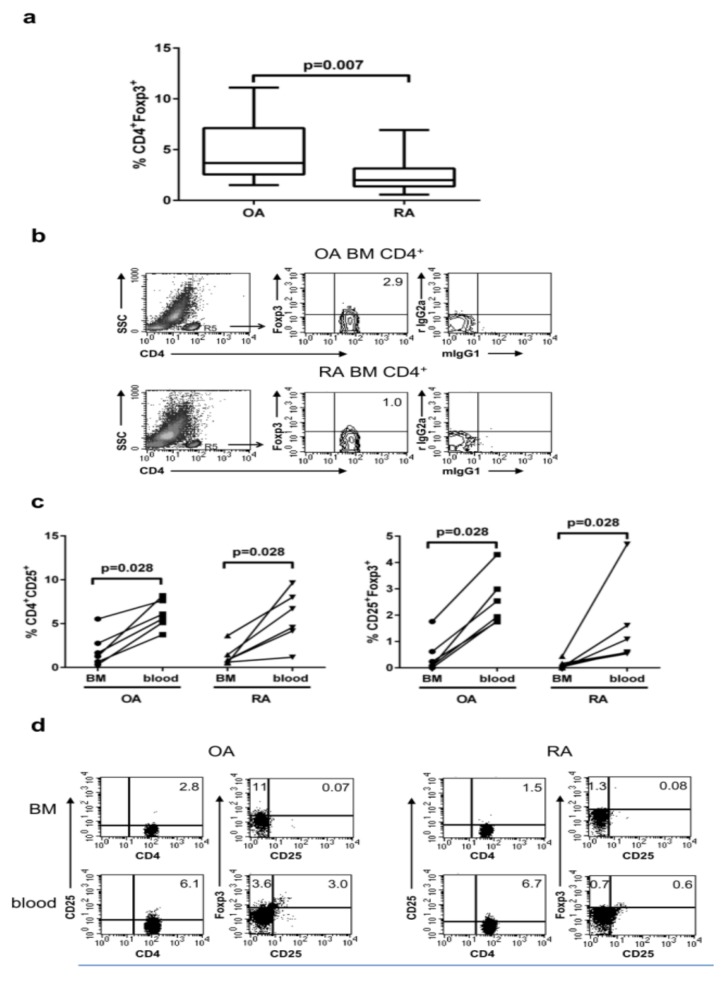
Analysis of CD4^+^FOXP3^+^ T cells population in the BM. (**a**) Proportions of CD4^+^FOXP3^+^ cells in the BM of OA and RA patients. Data are presented as median with a min–max range (*n* = 16 subjects per group). Differences between groups of patients were analyzed by Mann–Whitney U-test. (**b**) Representative dot plots show FOXP3 expression by gated CD4^+^ T cells in OA and RA BM, respectively. **(c)** The percentage of CD4^+^CD25^+^ and CD25^+^FOXP3^+^ among CD4^+^ T cells from the peripheral blood and BM of the same patient is shown (*n* = 6). (**d**) Representative dot plot show CD25 and FOXP3 expression by gated CD4^+^ cells in the BM and peripheral blood of the same patient. Comparison of the BM with the blood from the same patient (done separately for OA and RA patients) was analyzed by the Wilcoxon test. Numbers depicted on dot plots show the frequencies of subset expressing the proper marker. OA/RA BM/blood — cells isolated from the BM/peripheral blood of patients with OA/RA, respectively.

**Figure 3 cells-09-00549-f003:**
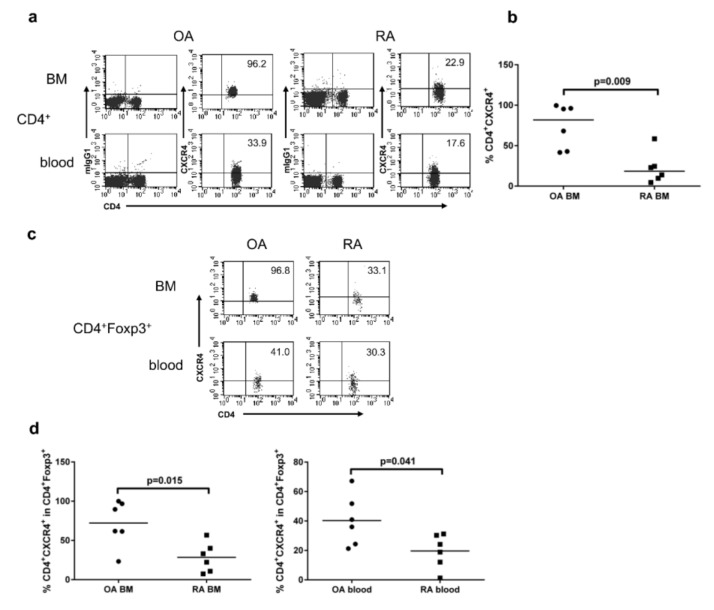
Expression of CXCR4 by CD4^+^ and CD4^+^FOXP3^+^ cells from the bone marrow and peripheral blood of OA and RA patients. (**a**) Representative dot plots show CXCR4 expression by gated CD4^+^ cells in the BM and peripheral blood of the same patient. (**b**) CXCR4 expression by CD4^+^ T cells in OA and RA BM. (**c**) Representative dot plots show CXCR4 expression by gated CD4^+^FOXP3^+^ regulatory T cells (Tregs) in the BM and peripheral blood of the same patient. (**d**) CXCR4 expression by gated CD4^+^FOXP3^+^ Tregs in the BM (chart on the left) and peripheral blood (chart on the right) of OA and RA patients. In all the cases, differences between groups of patients were analyzed by Mann–Whitney U-test. Numbers depicted on dot plots show the frequencies of subset expressing CXCR4. Individual results are shown as dots and median as a bar on charts (*n* = 6 subjects per group). OA/RA BM/blood—cells isolated from the BM/peripheral blood of patients with OA/RA, respectively.

**Figure 4 cells-09-00549-f004:**
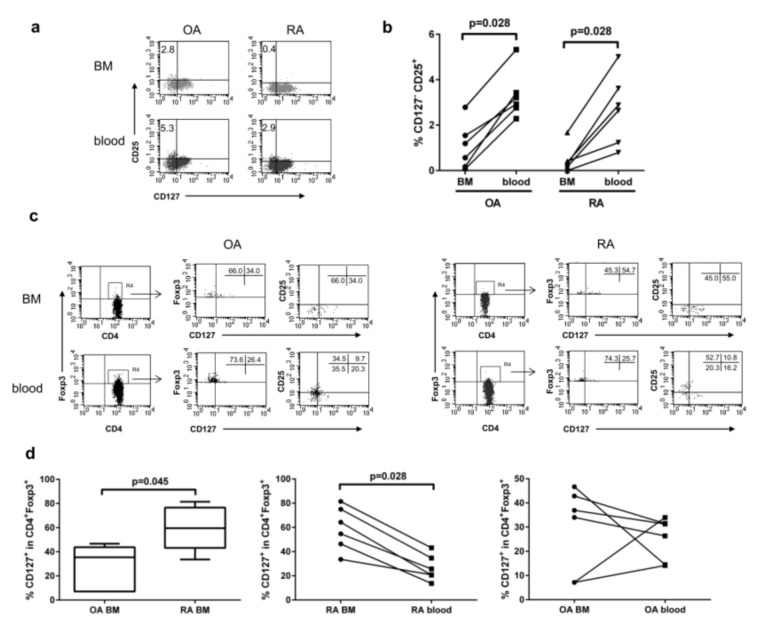
Expression of CD127 by CD4^+^FOXP3^+^ cells from the bone marrow and peripheral blood of OA and RA patients. (**a**) Representative dot plots show the expression of CD127 and CD25 by gated CD4^+^ T cells in OA and RA BM and peripheral blood. (**b**) The percentage of CD127^−^CD25^+^ among CD4^+^ T cells from the BM and peripheral blood of the same patient (*n* = 6). Individual results are shown as dots. Comparison of the BM with the blood from the same patient (done separately for OA and RA patients) was analyzed by the Wilcoxon test. (**c**) Representative dot plots show CD127 and CD127 together with CD25 expression by gated CD4^+^FOXP3^+^ cells in the BM and peripheral blood of OA and RA patients. (**d**) The percentage of CD127^+^ cells among CD4^+^FOXP3^+^ T cells from the BM of OA and RA patients (chart on the left), and the BM and peripheral blood of the same patient (RA—chart in the middle, OA—chart on the right; *n* = 6). Data on the left chart are presented as median with a min–max range, and differences between groups of patients were analyzed by Mann–Whitney U-test. On the remaining two charts, individual results are shown as dots, and a comparison of the BM with the blood from the same patient was analyzed by the Wilcoxon test (done separately for OA and RA patients). Numbers depicted on dot plots show the frequencies of subset expressing a proper marker. OA/RA BM/blood—cells isolated from the BM/peripheral blood of patients with OA/RA, respectively.

**Figure 5 cells-09-00549-f005:**
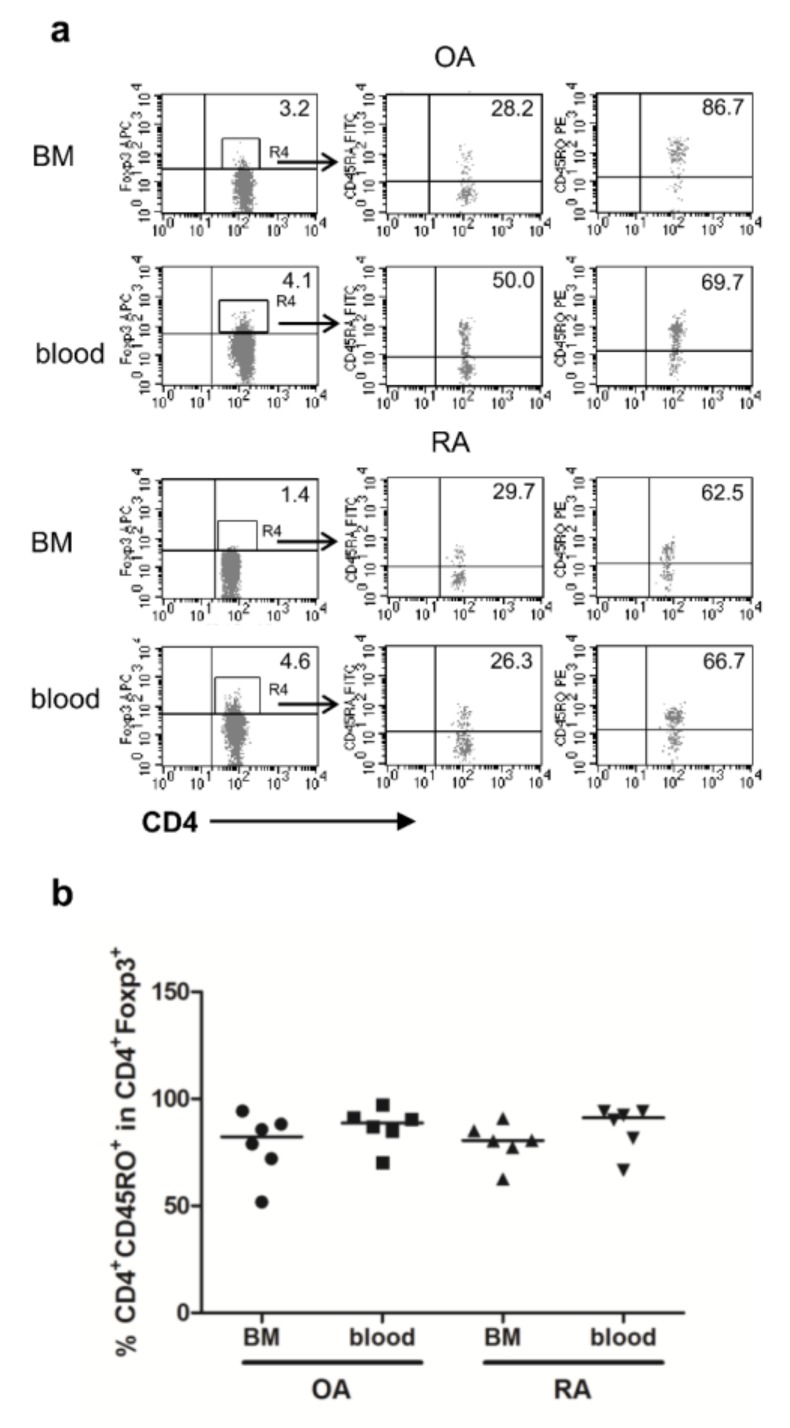
CD45RO expression by gated CD4^+^FOXP3^+^ cells in the BM and peripheral blood of OA and RA patients. (**a**) Representative dot plots show CD45RA and CD45RO staining of CD4^+^FOXP3^+^ cells from the BM and peripheral blood from the same patient. (**b**) Expression of CD45RO by CD4^+^FOXP3^+^ Tregs from the BM and peripheral blood from OA and RA patients. Individual results are shown as dots and median as a bar on charts (*n* = 6 subjects per group). OA/RA BM/blood—cells isolated from the BM/peripheral blood of patients with OA/RA, respectively.

**Figure 6 cells-09-00549-f006:**
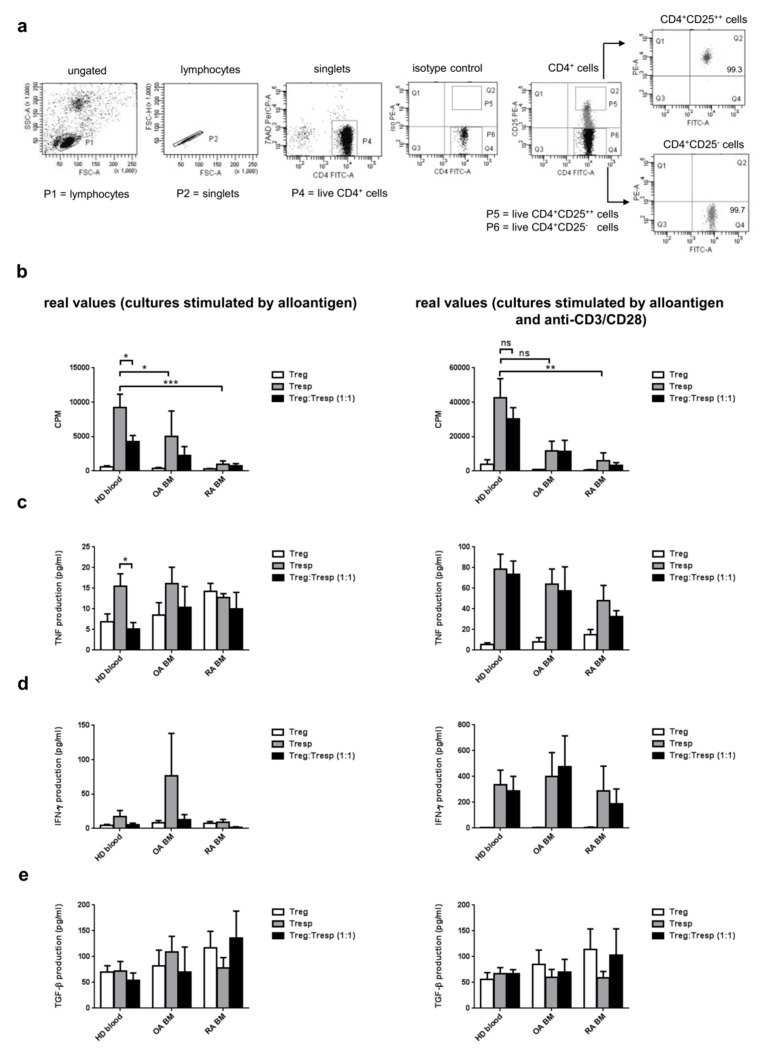
The suppressive potential of BM Tregs. (**a**) Representative sorting strategy for CD4^+^CD25^++^ and CD4^+^CD25^−^ subsets from the BM and peripheral blood (here OA BM is shown). (**b**) Proliferation of Tregs, responder T cells (Tresps) and their coculture is shown as real values (cpm). CD4^+^CD25^++^ Treg and CD4^+^CD25^−^ Tresp cells were isolated from the peripheral blood of healthy donors (*n* = 10) and the BM of OA (*n* = 7) and RA (*n* = 8) patients. Detection of TNF (**c**) IFN-γ (**d**) and TGF-β (**e**) produced by Tregs, Tresps or their coculture is shown as pg/mL. Results were obtained from cultures of cells isolated from the peripheral blood of healthy donors (*n* = 9) and the BM of OA (*n* = 6) and RA (*n* = 6) patients. (**b**–**e**) Statistical significance was calculated comparing results for Tresp with those obtained for Tresp cultured together with Treg. The differences between the groups were tested for their statistical significance using parametric two-tailed T-test or Mann–Whitney U-test when the normality assumption was not met (GraphPad Software, USA). The results are shown as mean ± SEM. **p* < 0.05, ***p* < 0.01, ****p* < 0.001; ns—not significant. HD blood—cells isolated from the peripheral blood of healthy donors; OA/RA BM—cells isolated from the BM of patients with OA/RA, respectively.

**Table 1 cells-09-00549-t001:** Summary of clinical data of patients included in the study.

Diagnosis	OA (*n* = 42)	RA (*n* = 36)
Age (mean years ± SEM), (range)	57.3 ± 1.3 (34–70)	51.8 ± 2 (24–71)
Gender	29F/13M	26F/10M
ESR (range)	13 ± 1.2 (2–37)	32.6 ± 2.8 (3–58)
Disease duration (mean years ± SEM), (range)	Data not available	20.6 ± 1.7 (7–50)
**Treatment**		
**NSAID:**		
Meloxicam	1	N/A
Diclofenac	13	19
Ketoprofen	2	N/A
Nimesulid	1	N/A
Naproxen	1	N/A
**Not-NSAID:**		
Paracetamol	8	N/A
Tramadol	4	N/A
Paracetamol + Tramadol	1	1
**DMARD:**		
Methotrexate	N/A	11
Sulfasalazine	N/A	3
Leflunomide	N/A	2
Azathioprine	N/A	1
**Steroids:**		
Methylprednisolone	N/A	21
Encorton	N/A	9

**Table 2 cells-09-00549-t002:** Percentage of CD4^+^FOXP3^+^ cells in CD4^+^CD127^+/−^CD25^+/−^ subsets in BM mononuclear cells (BMMCs) and peripheral blood mononuclear cells (PBMCs) isolated from OA and RA patients (*n* = 6 in each group); ns—not significant.

**OA BMMCs**				**OA PBMCs**			
	**Mean % of the Cells**	**Range**	**Median**	**Mean % of the Cells**	**Range**	**Median**	***p*** **-Value**
CD127^+^CD25^+^	4.8	0.0–14.3	1.8	21.0	5.0–36.4	21.9	0.02
CD127^+^CD25^−^	1.8	0.2–5.6	1.3	1.1	0.2–1.9	1.1	ns
CD127^–^CD25^+^	41.0	14.3–62.5	47.1	59.4	43.7–75.0	57.6	ns
CD127^–^CD25^−^	11.9	5.2–38.5	6.8	8.9	5.1–13.3	8.5	ns
**RA BMMCs**				**RA PBMCs**			
	**Mean % of the Cells**	**Range**	**Median**	**Mean % of the Cells**	**Range**	**Median**	***p*** **-Value**
CD127^+^CD25^+^	7.7	0.0–20.0	3.9	13.6	2.2–33.3	8.3	ns
CD127^+^CD25^−^	1.6	0.9–2.5	1.4	0.9	0.2–2.1	0.8	ns
CD127^−^CD25^+^	9.3	0.0–28.6	3.6	41.7	13.5–77.7	40.2	0.016
CD127^−^CD25^−^	4.5	1.7–11.8	2.5	8.4	2.6–22.5	4.3	ns

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
