# Peer review of "CD4+FOXP3+ T Cells in Rheumatoid Arthritis Bone Marrow Are Partially Impaired"

_cells, 2020, doi:10.3390/cells9030549_

Round 1

Reviewer 1 Report

In the reviewed manuscript the authors present data on the presence and functionality of regulatory T cells (Tregs) within bone marrow of patients with rheumatoid arthritis (RA) and osteoarthritis (OA). Massalska et al. compare the number and phenotype of Tregs residing within the bone marrow with the phenotype of circulatory Tregs isolated from the peripheral blood in both diseases. Additionally, authors test the suppressive activity of Tregs from the studied patients.

Although the manuscript is generally interesting, there are several points and shortfalls which need addressing:

Major points:

Patients numbers are really low (n=6) for most of the figures, but in the material and methods authors state that 36 patients in RA group and 42 patients in OA group were studied. That requires an explanation. FoxP3 flow cytometry staining needs improvement. I can accept that it might be difficult to get nice staining from the bone marrow, but staining from the blood is equally unconvincing though out the manuscript Fig 2 – can author include a FACS plot for CD25 staining and CD25hi cells gating, please Fig 4d – again gating is unconvincing, in C could authors gate CD4+FoxP3+ as shown and then CD25 vs CD127, please Fig 6 - suppression tests – it is commonly accepted in the field that positive control cpm values should be above 10,000 cpm to make tests interpretable. Therefore, it seems from the data presented that data for RA samples can not be interpreted, due to the low proliferation of stimulated positive control samples. Additionally, please change – cultures with authors call “not stimulated cultures” are in fact alloantigen stimulated (by accessory cells). Also B-E please reanage data on the graphs to show samples from the same patient group next to each other – the main comparison is within patient group (do Tregs suppress as compared to non-Treg control) and it should be reflected on the graphs.

Minor points:

P2V48-49 (page 2 verse 48-49) – new classification recommends term peripherally-induced Tregs (pTregs) instead of induced Tregs (iTregs) and thymic Tregs (tTregs) instead of nTregs. pTregs are derived from naïve T cells. Please correct Please read carefully to check for grammatic errors like f.e. p2v60 “several findings has” V94-95 – please state which immunosupressive drugs Figure 1 – please add information about how many patients had histology staining done

Reviewer 2 Report

The Authors provide an extensive analysis of bone marrow and peripheral regulatory T cells expression in patients affected by rheumatoid arthritis using as a control group patients affected by osteoarthritis and peripheral blood samples from healthy donors.
The study appears well conducted, and the Authors discuss and analyze the results extensively.
Minor issues:
Line 87: The Authors explain the meaning of OA, but they have introduced this abbreviation before (line 78).
Line 92: Could the Authors also indicate the disease duration range?
Line 94: Could the Authors indicate if any patient was treated with biological DMARDs?
Line 259: The Authors could substitute "with the similar" with "with a similar".
Line 261: The Authors could substitute ", and only the tendency was observed" with ", and only a tendency was observed".
Lines 339-345: Do the Authors analyzed the suppressive activity of Tregs from peripheral blood of RA and OA patients? Was it higher than the suppressive activity of Tregs from BM?
Line 350 and following: correct IFN-@ with IFN-γ
Lines 370-371: What do the Authors mean with "may cause the reason why the chronic diseases are still incurable today"? Which chronic diseases do they mean? Autoimmune diseases? Rheumatoid arthritis? However these diseases can be adequately cured using modern approaches including biological DMARDs. The Authors must rephrase and better explain this concept.

Reviewer 3 Report

Massalska et al. show that Treg (CD4+ FoxP3+ CD25+ T-cells) isolated from the bone marrow (BM), but not blood, are impaired in their suppressive ability when test in vitro.  BM cells were obtained from individuals with RA or OA, undergoing hip replacement surgery.

This an interesting manuscript that makes an important contribution to the field. There are minor concerns and some suggestions that would improve the manuscript.  

The entire manuscript needs to be edited to improve grammar and phrasing. The  For instance, line 214 (Results heading) reads "RA BM Treg exert low expression of CXCR4" should read "Low expression of CXCR4 is observed in RA BM Treg" or something like it.  Cells do not exert expression of genes.   Figure 1 does not add much to the overall story here. It could be moved to Supplementary figure.  In any case, please add scale bars to the images.   Can the authors provide an age of individual vs. BM Treg levels, if there is enough spread in age data?  The author should report on CCR7 expression on the CD5RO+ cells, if possible.  

Reviewer 4 Report

The manuscript of Massalska et al. claim that BM Tregs from RA patients as compared with OA patients have lower frequency and limited suppressive activity on conventional T cells. However, according to the way analyses were performed and the technical flaws in data presentation, their conclusions are not convincing.
Major concerns: 1. Groups of RA patients receiving different drugs are analyzed, but no correlation with the type of drug and its potential effect on the number and function of BM Tregs has been considered. In the OA group of patients which are supposed to be the control group, all have been treated with non-steroid drugs, whilst the RA patients received 7 different drugs before surgery. The type of drugs in each analyzed group of RA patients (based on the type of drug received) is not provided, neither discussed.
2. Data in all figures are meaningless as long as the type of patients based on drug treatment are presented. What group of patients each dot on the graphs represent?
3. The CD127 marker of central memory T cells is inappropriately associated with activation
4. Comparison of Tregs from BM vs. synovial liquid at the time of surgery is totally missing.
5. Identification of Tregs by FACS relies solely on the CD25 and no intracellular staining for is shown.
Minor: FOXP3 spells for human Tregs, not Foxp3.
If cells were gated on FOXP3+CD4+ or CD4 only, why CD4 marker is represented on all FACS quadrants? This is really confusing (see fig.3c for example).

Round 2

Reviewer 4 Report

The manuscript of Massalska et al. has been significantly improved. However, there are several minor revisions required: 1. Add the word “cells” or “Tregs” following CD4 FOXp3+….throughout the manuscript (ex: see lanes 312, 314. 322, 328, 413, etc.) 2. In Fig. S1 replace CD25++ with CD25high 3. 3. A list of all used abbreviations needs to be added to the manuscript.
